# Was Drought Really the Trigger Behind the Syrian Civil War in 2011?

**Arnon Karnieli [1],\*, Alexandra Shtein [1], Natalya Panov [1], Noam Weisbrod [1] and Alon Tal [2]** 

[1]   Jacob Blaustein Institutes for Desert Research, Ben-Gurion University of the Negev, Sede Boker Campus, Sede Boker 84990, Israel

[2]   Department of Public Policy, Tel Aviv University, Tel Aviv 69978, Israel

\*   Correspondence: karnieli@bgu.ac.il; Tel.: +972-52-879-5925

**Abstract:** The role played by unsustainable resource management in initiating international conflicts is well documented. The Syrian Civil War, commencing in March 2011, presents such a case. The prevailing opinion links the unrest with sequential droughts occurring from 2007–2010. Our research, however, reveals that the winter-rainfed agricultural conditions before 2011, as detected by satellite-derived vegetation indices, were similar and even better for Syrian farmers than for those of their Turkish counterparts across the border. Concurrently, summer-irrigated crops, heavily dependent on Euphrates River water originating from Turkey, notably declined in Syria while flourishing in Turkey. These findings are firmly supported by other independent and validated datasets, including long-term cross-border discharge, the water level in Syrian and Turkish reservoirs, and transborder groundwater flow. We conclude that the Turkish policy of unilaterally diverting the Euphrates water was the main reason for the agricultural collapse and subsequent instability in Syria in 2011. The obvious inference is that while prolonged drought exacerbated conditions, unsustainable anthropogenic water management in Turkey was the proximate cause behind the Syrian uprising.

**Keywords:** Syrian Civil War; Euphrates; Turkish Southeastern Anatolia Project; Normalized Difference Vegetation Index (NDVI); discharge; reservoir water level; transborder groundwater flow

## 1. Introduction

Several recent publications, e.g., [1–6], and innumerable commentators, have posited that consecutive droughts and climate change were the primary causes of the 2011 unrest in Syria and the subsequent military conflict. According to this view, the breakdown of Syrian agriculture caused mass rural-urban migration, economic turbulence, and civil instability. We challenge these claims based on an analysis of spaceborne-derived vegetation signals and hydrological data before and during the Syrian conflict (2011–2015). We argue that the main reason for the widespread agricultural failure in Syria from 2010–2011 involved the recent buildup of water reservoirs in the upper Euphrates Basin, which allowed Turkey to expand its utilization and divert the historical flow into Syria. The reduction in the availability of this transboundary surface and underground water constitutes an alternative, more compelling, proximate cause of the irrigated agricultural decline that drove the population exodus from northern Syria's rural regions.

In the vicinity of their shared border, Turkey and Syria are not separated by meaningful geographical, climatic, or agricultural circumstances. The common climatic conditions are semiarid, characterized by precipitation ranges averaging from 200–350 mm yr$^{-1}$, between November and April, with high yearly variations [7]. Despite the pervasive erratic geopolitical and seasonal dynamics in the region, the land in and around the Euphrates and Tigris Rivers, part of the ancient Fertile Crescent, has historically provided sufficient water and nutrients for sustainable agriculture [8]. The Euphrates and its

three main tributaries—the Sajur, Balikh, and Khabour—originate from precipitation or groundwater in the mountains of eastern Turkey (Figure 1). These rivers contribute about 90% of the unified Euphrates flow and provide the main sources of irrigation water in northeastern Syria. More than 70% of the river water is used for irrigation while the remainder produces hydroelectric power and drinking water [9]. With 60% of its water resources originating outside its borders, particularly in the Euphrates Basin, Syria has always been aware of its vulnerability to upstream extractions [10]. Over the years, it pursued a series of agreements with neighboring Turkey to ensure access to waters from the watershed, so critical to the irrigation of local farms [9]. Among these, the first interim agreement between Turkey and Syria, signed in 1987, states that 16,000 million m$^3$ of water will be released annually across the border. The complementary 2009 agreement addressed common hydrological initiatives, including the construction of water pumping stations and shared dams, as well as the development of a joint water policy.

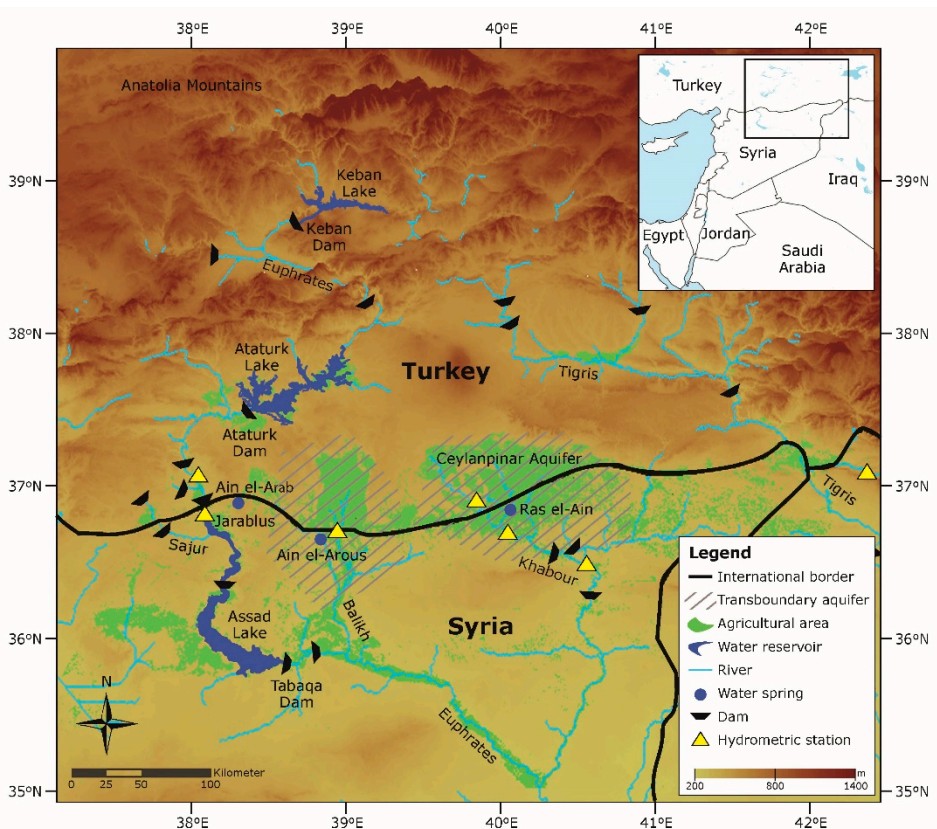

**Figure 1.** The Euphrates River Basin with its water structures, including the main dams, water reservoirs, and hydrometric stations (based on data from [9]).

Reports prior to 1938 measured the natural annual flow of the Euphrates at the hydrometric station near the town of Jarablus (located on the Syrian-Turkish border) at 1200 m$^3$ s$^{-1}$ (Figure 2) [9]. Years with low annual flows during the early 1970s and early 1990s can be attributed to the diversions that occurred while new water reservoirs in Turkey (Keban and Ataturk) were being filled. Still, data from the last 70 years reveal a statistically significant negative trend in flow, indicating a decrease in the mean annual discharge of about 500 m$^3$ s$^{-1}$ in 2010. To some degree, this is due to climate variability and more frequent drought periods. However, a more significant contribution to the reduced flow is the ambitious Turkish Southeastern Anatolia Project (GAP): 22 large dams and 19 hydroelectric power stations on the rivers in the upper Euphrates and Tigris Basins, supported by an infrastructure designed to store water during the rainy season and release it during the dry months [11,12].

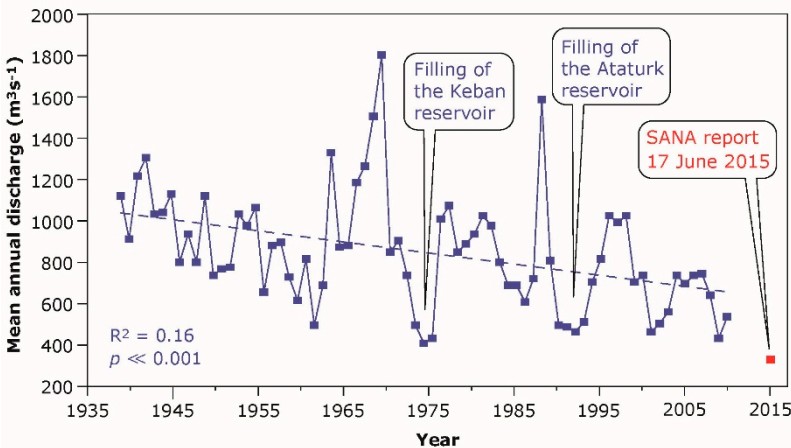

**Figure 2.** Mean annual discharge of the Euphrates River as monitored at the Jarablus hydrometric station, revealing a statistically significant negative trend until 2010 (based on data from [9]). The extremely low value for 2015 was reported in [13].

The project was gradually extended, beginning in the late 1990s, with the objective of providing irrigation to 1800 km$^2$ of land. The 14 Turkish reservoirs on the Euphrates and its tributaries, with a maximum storage capacity exceeding $144 \times 10^9$ m$^3$, significantly affected the natural flow regime in the river during recent decades. No data are available from the Jarablus station after the Syrian unrest in 2011, when the area was conquered by the Islamic State of Iraq and Syria (ISIS) forces, as shown in Figure 3.

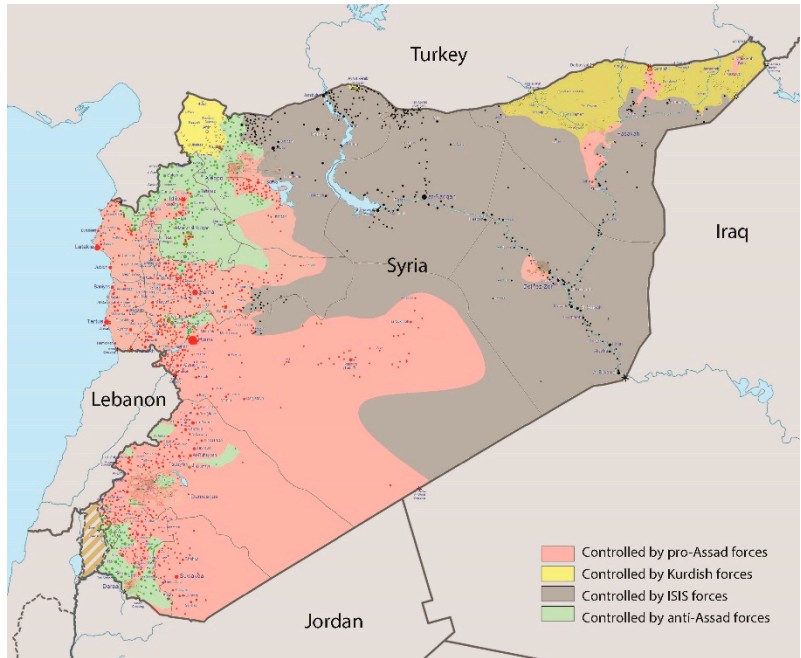

**Figure 3.** The territorial control map of the Syrian Civil War as of 24 November 2014 (reproduced from Wikimedia Commons, https://upload.wikimedia.org/wikipedia/commons/6/60/Syrian_civil_war_24-11-14.png).

Nonetheless, on 17 June 2015, Syrian officials reported that the water discharge passing the Jarablus station was as low as 330 m$^3$ s$^{-1}$ during the first half of 2015 [13]. Additionally, as a result of the accelerated agricultural development in southern Turkey, the surface and subsurface water containing pesticides and fertilizers that left the irrigated fields, termed "return flow", degraded the water quality downstream. Reports described a gradual increase in salinity levels when the water

entered Syria [14,15]. Although the overall discharge of the three main tributaries, the Sajur, Balikh, and Khabour, was only 8% of the Euphrates, their flows also dramatically decreased over the years [15].

## 2. Materials and Methods

### 2.1. Vegetation Index Time Series

A long-term (33 years) Normalized Difference Vegetation Index NDVI time series was created from two continual satellite datasets: the Advanced Very High Resolution Radiometer onboard the National Oceanic and Atmospheric Administration (NOAA-AVHRR) satellite from 1982 to 1999 (AVH13C1 product, http://ltdr.nascom.nasa.gov/ltdr/productSearch.html) and the Moderate Resolution Imaging Spectroradiometer (MODIS) from 2000 to 2015 (MOD13Q1 product, http://reverb.echo.nasa.gov/reverb). To demonstrate the peak winter and summer growing seasons, the data from both sources were retrieved for two 16 day periods: the end of the winter season (5–20 March) and the end of summer (28 August–12 September).

The two products differ in their spatial and temporal resolutions. Therefore, different processing steps were applied to each of them. The temporal resolution of the AVH13C1 product is daily, and the spatial resolution is 1.1 km. The preprocessing steps of this product included: importing Hierarchical Data Format (HDF) files into the image IMG format, cropping the image into the area of interest, stacking all images together, and applying a pixel quality mask to exclude cloudy, shaded, and invalid pixels. For each year, the 16 daily images were aggregated into one maximum value composite image. Subsequently, all years were stacked into one image that was divided by a scale factor and resampled to 250 m for matching the pixel size of the MODIS product.

The temporal resolution of the MOD13Q1 product is 16 days, and its spatial resolution is 250 m. The algorithm of this product chooses the best available pixel value from all the acquisitions throughout the 16 day period. The criteria used involved low clouds, a low view angle, and the highest NDVI value. The preprocessing steps of this product included: importing HDF files into the IMG format, cropping the image into the area of interest, dividing the product by a scale factor, and re-projecting to a geographic coordinate system.

A masking operation was performed by the MODIS NDVI product for discriminating the agricultural fields from the surrounding bare areas in the analysis. This procedure was carried out by matching the criteria of the maximum NDVI value composite of the 16 years (2000–2015) of summer images and a threshold of NDVI > 0.2. Then, all agricultural areas in the image were converted into polygons. Only major polygons were selected and linked separately to the Syrian and Turkish territories. Finally, these polygons were applied to the NDVI time series.

Before calculating the statistical measures, all NDVI pixels with values ≤0.0 were assigned as "no data" pixels to avoid considering water bodies in the analysis. For each country, the average and the standard deviation statistics of the agricultural areas were calculated for the two seasons throughout the 33 years of observations.

The image processing procedures were implemented with ERDAS Imagine (https://www.hexagongeospatial.com/products/power-portfolio/erdas-imagine) and ArcGIS 10.1 software packages (ESRI, https://www.esri.com/en-us/home).

### 2.2. Assessment of Magnitude of Change

The difference between NDVI over the Turkish and Syrian territories and its change over the years was quantified with the Cohen's D Effect Size Index (D) for differences between the means [16,17]. While hypothesis-based test statistics, e.g., Student's *t*-test, assign a significance level to the observed difference, the D statistics estimates the strength, or the magnitude, of the difference. As such, it avoids using the sample size, thus avoiding false positive results for large samples. Thus, D constitutes a better measure of the difference between the means of the large samples (number of analyzed pixels in Turkey and Syria, 122,045 and 111,511, respectively).

The index is computed as:

$$D = \frac{\overline{X_1} - \overline{X_2}}{s} \tag{1}$$

where $\overline{X_1}$ and $\overline{X_2}$ are the two means, and the pooled standard deviation (s) is computed as:

$$s = \sqrt{\frac{(n_1 - 1)s^2{}_1 + (n_2 - 1)s^2{}_2}{n_1 + n_2 - 2}} \tag{2}$$

where $s^2{}_1$ and $s^2{}_2$ are the variances of the two samples. As the difference between the two samples increases, the value of D increases, where D = 0 indicates that the means of the two samples are equal, D = 0.2 defines a small difference, D = 0.5 a medium difference, D = 0.8 a large difference, and D = 1.2 indicates a very large difference [17].

*2.3. Water Level in Reservoirs*

The time series of respective water levels were derived from the Database for Hydrological Time Series over Inland Waters (DAHITI) relying on an extended outlier rejection and a Kalman filter approach incorporating cross-calibrated multi-mission altimeter data from Envisat, ERS-2, Jason-1, Jason-2, TOPEX/Poseidon, and SARAL/AltiKa spacecrafts (http://dahiti.dgfi.tum.de/en/) [18].

## 3. Results

*3.1. Temporal Dynamics of Agricultural Areas*

Spaceborne remote sensing provides periodic and repetitive digital data on regional and global scales, generating critical information about sites that are inaccessible or unmonitored due to extreme conditions. It also provides a historical record and long-term observations, revealing causes and effects of climate and/or human-induced changes on the ground that are neither reported nor measured directly. A useful indicator of environmental change is the Normalized Difference Vegetation Index (NDVI) that enhances the vegetation signal and makes it possible to distinguish between vegetation states [19]. Dense and healthy vegetation has relatively high NDVI values that contrast with soil's low NDVI values.

A long-term (1982–2015) NDVI time series reveals that the annual mean NDVI values for September increased, in parallel, in Turkey and Syria until 2009, reflecting the steady growth of irrigated agriculture in both countries (Figure 4). Then in 2010, the trends change: a significant divergence can be seen with a sharp increase in NDVI values in Turkey, indicating expanded agricultural activities, in contrast to noticeable decreases in Syria. Similar trends were detected by processing a time series of Landsat images during the same period.

Cohen's D Effect Size Index (Figure 5) quantifies the difference between the mean NDVI values in Turkey and Syria (shown in Figure 4). While until 2011, the magnitude of D fluctuated between −0.5 and 0.5, a progressive increase has been apparent ever since 2011. In 2015, D = 1.25 was categorized as a very significant difference [16,17], indicating that the mean NDVI in Turkey was more than 1 (pooled) standard deviation greater than the mean NDVI in Syria.

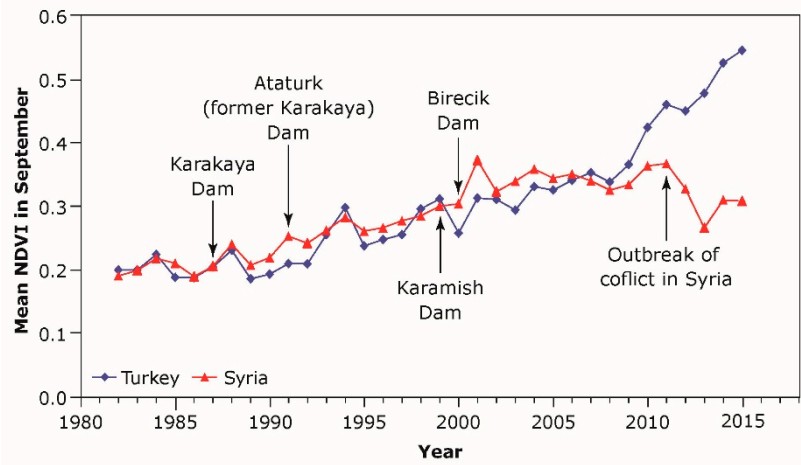

**Figure 4.** Long-term time series of the mean Normalized Difference Vegetation Index (NDVI) in September. Values gradually increased in both Turkey and Syria until 2008, reflecting steady agricultural growth in both countries. Later, NDVI values dramatically increased in Turkey and sharply decreased in Syria. Values were extracted from a combined dataset of NOAA-AVHRR and MODIS spaceborne systems.

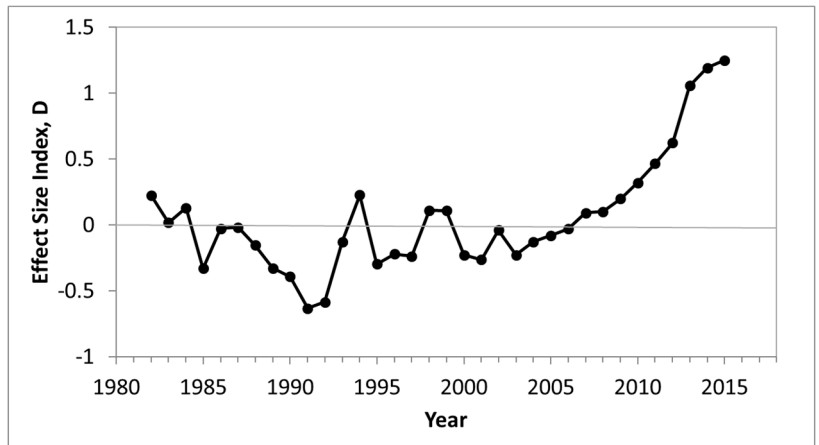

**Figure 5.** The temporal trend of the Cohen's D Effect Size Index for the difference between the mean NDVI values in Turkey and Syria (Figure 4). Until 2011, the magnitude of D was considerably small but has gradually increased since 2011. In 2015, the index was categorized as a very large difference.

Given the Mediterranean climate, March is the end of the region's rainy season. With the heading stage of cereals, in March 2011, the vegetation signal (in terms of departure from long-term mean NDVI values) appears slightly higher in Syria than in Turkey (Figure 6A). This finding is consistent with international reports [9,20] of wheat production in 2011 being 13% higher than the previous five year average due to improved seasonal rainfall and temperature conditions. This suggests that rainfed agriculture was unaffected by the drop in the availability of Euphrates water. Furthermore, no notable difference between the two countries was observed in March 2015 (Figure 6B). In September 2011, however, just before the advent of the rainy season when farmers are most dependent on irrigation, vegetation signals observed in Turkey are significantly higher than in Syria (Figure 6C). This clear contrast becomes even more pronounced in September 2015 (Figure 6D). Turkish crops exhibit bounteous conditions, while Syrian agriculture hardly exists because of the inadequate quantities of water flowing across the border.

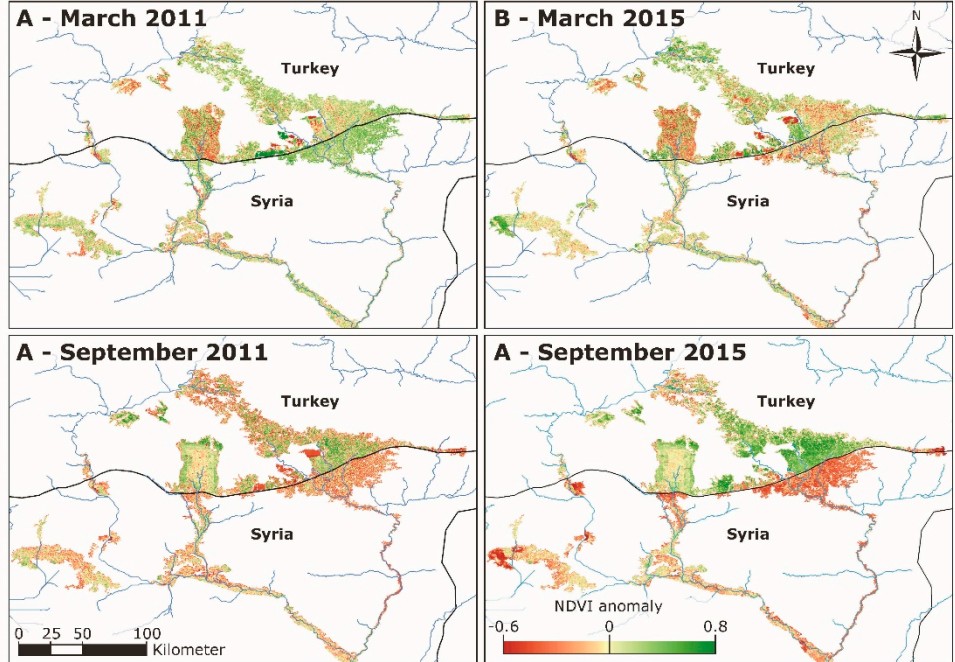

**Figure 6.** The MODIS-derived NDVI anomaly (departure from the normal) for key months in Turkey and Syria. March 2011 (**A**); March 2015 (**B**); September 2011 (**C**); and September 2015 (**D**). The March images, which represent rainfed crops, do not show a significant NDVI.

*3.2. Water Level in Turkish and Syrian Reservoirs*

Corroborating evidence emerges from the two main water reservoirs that store Euphrates water for irrigation. First, Ataturk Lake in Turkey was extended over an area of 817 km$^2$, with the water volume reaching 48.7 km$^3$, enabling irrigation of nearly 4760 km$^2$ of arable land as part of the GAP. The other sign involves Syria's Assad Lake, with a surface area of 610 km$^2$ and a capacity of 11.7 km$^3$, designed to irrigate an area of 6400 km$^2$. The comparison shows a dramatic increase in Ataturk Lake volume during 2010, with a corresponding progressive drop in the volume of Assad Lake (Figure 7).

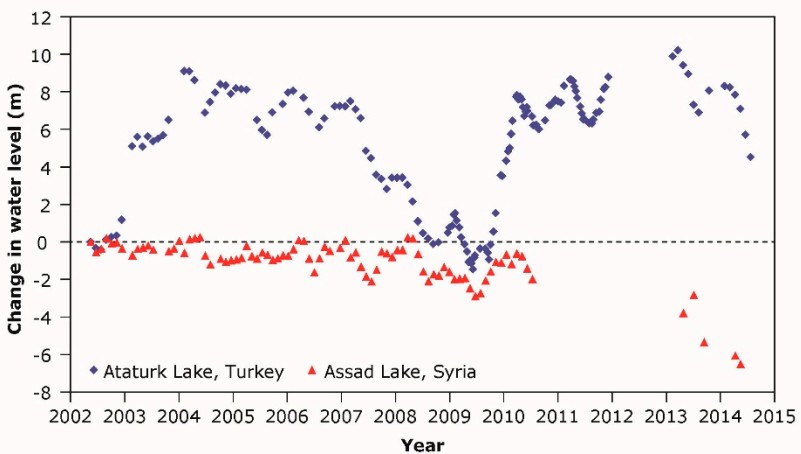

**Figure 7.** The temporal change of water levels in the Ataturk Lake (Turkey) and Assad Lake (Syria) relative to July 2002. Both reservoirs show a stable level until 2011 (except for the gap in Ataturk Lake between 2008 and 2010), while the water level in the Syrian lake gradually decreased after 2013. Data extracted from several space missions (http://dahiti.dgfi.tum.de/en/).

*3.3. Transborder Groundwater Flow*

Precipitation falling over the southeastern Antalya Mountains recharges the Turkish–Syrian transboundary aquifers (Figure 1). For thousands of years, springs and water wells were fed by shallow groundwater and used for irrigation. During the early 1960s, a cluster of 30 springs, named Ras el-Ain, within the Ceylanpinar Aquifer (ca. 2000 km$^2$), abounded with a flow of about 40 m$^3$ s$^{-1}$ and were considered to be among the largest karst springs in the world [21]. Similarly, smaller water springs, such as the Ain el-Arous, the main source of the Balikh tributary, and Ain el-Arab produced a mere 7 m$^3$ s$^{-1}$ in 1980 [22]. Throughout the subsequent decades, however, a vast expansion of the cultivated areas on both sides of the border took place. In Turkey, the process was intentionally engineered by the government as the groundwater-irrigated component of the GAP [23]. This accelerated extraction process was also driven by the broad introduction of diesel-powered pumps, enabling abstraction of water at depths of tens to hundreds of meters belowground [24]. In response to the flow shortages in the tributaries of the Euphrates River, Syrian farmers were forced to drill more numerous and deeper wells to meet existing consumption needs. In 2001, the World Bank reported that some 1550 privately developed wells were being utilized in the region, most of them unlicensed or "pirate" in nature, managed irrationally and unsustainably [25]. Gradually, agricultural operations grew increasingly dependent on groundwater. Data from the Gravity Recovery and Climate Experiment (GRACE) satellites [26] were used for calculating the freshwater storage trends in the north-central Middle East, including portions of the Euphrates and Tigris River Basins [27]. Analyses show that the monthly groundwater storage variations in the region, presented as a deviation from the normal, were almost stable (4.9 ± 3.1 mm yr$^{-1}$) between January 2003 and December 2006, but sharply declined (−34.0 ± 4.5 mm yr$^{-1}$) between January 2007 and December 2009. The resulting overexploitation of groundwater caused a dramatic depletion of the water table in many areas across Syria, including its northern districts [22,28]. The flow discharge of the Ras el-Ain Springs declined to only a few m$^3$ s$^{-1}$ and ultimately has disappeared completely since 2001 [22,25]. Correspondingly, Ain el-Arous and Ain el-Arab have totally dried up since 1985. It was predicted that if overutilization continues, the aquifers' resources would not be able to recover within a reasonable timeframe [23].

*3.4. Winter vs. Summer Crop Production*

During the rainy season, the main winter-rainfed crops in Syria and Turkey are cereals (wheat and barley) [29]. About half the Syrian grain production was traditionally cultivated in this border region. During the summer, cotton and, to some extent, fruits and vegetables are dependent on irrigation water. Cotton remains Syria's most important cash crop, supplying the domestic textile industry, as well as the export market [29]. Our interpretation is further supported by an evaluation of the annual production of these two water-intensive crops (Figure 8). USDA sources (http://www.indexmundi.com/) report that wheat yields (representative of Syrian rainfed crops) were of average value in 2011, and only fluctuated modestly during the following years. In contrast, cotton, which represents locally irrigated crops dependent on the Euphrates' flow, has exhibited drastic production drops since 2011, vanishing almost entirely in 2016.

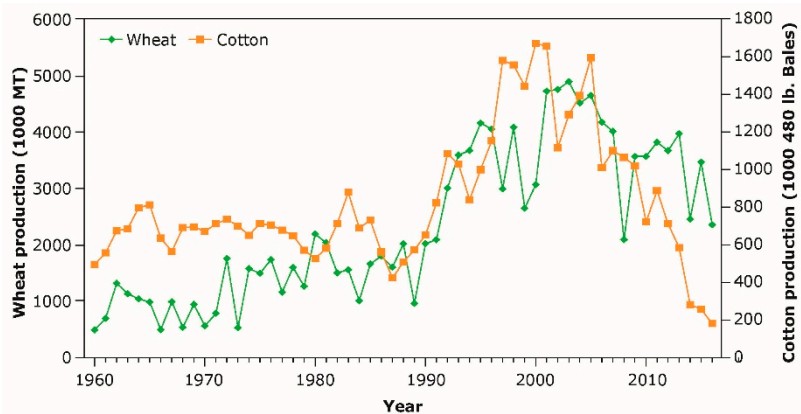

**Figure 8.** Temporal dynamics of wheat production, as representative of a winter-rainfed crop, and cotton production, as representative of a summer-irrigated crop. A severe production drop in the summer-irrigated crop is exhibited after 2010. Data extracted from http://www.indexmundi.com/.

## 4. Discussion

Climate change unquestionably has begun to affect precipitation patterns in the Mediterranean region where Syria is located [30,31]. However, spaceborne image processing suggests that the dramatic drop in agricultural yields and subsequent abandoning of farms that occurred in Syria in 2010 were driven by the decrease in available irrigation water, a result of substantial diversions in the Euphrates Basin initiated by Turkey. This situation was most likely brought about by the Turkish government's violation of its commitment to allow a minimum flow in the Euphrates at its Syrian border when Turkey pursued its ambitious reservoir strategy. Our synthesized research offers an alternative explanation to the prevailing narrative, which blames new levels of drought for the country's political instability, by showing that this narrative is inconsistent with actual environmental conditions. A range of quantifying factors, based on several independent sources, firmly point to reduced Euphrates flow as the most immediate cause of the 2011 Syrian agricultural collapse.

The United Nations projects that by 2025, roughly two-thirds of the people in the world will live under water-stressed conditions, and some 1.8 billion will live in countries with absolute water scarcity. The destabilizing and displacing potential of water shortages is increasingly recognized as climate change begins to transform precipitation patterns in drylands around the world. In the present case, unilateral water management decisions upstream were the primary driver of the massive crop failures, but Syria's experience also suggests that the protracted drought exacerbated the vulnerability of the country's local farmers. More precise characterization of the environmental factors behind the present refugee crises and their violent repercussions is critical if policymakers are to prevent similar natural resource-based conflicts and address the problem of climate refugees in the future. Unfortunately, short-term innovations by administrative levels, as proposed in other countries [32] for supporting change towards more sustainable and water-secure futures, do not seem to be effective in the current situation, since two countries are involved. Alternatively, international water-use treaties and the resolution of disagreements through international water law are proposed as strategies [27].

**Author Contributions:** Conceptualization, A.K. and A.T.; Methodology, A.K., N.P., and A.S.; Software, A.S., N.P.; Formal analysis, A.K., A.T., and N.W.; Writing—Original Draft Preparation, A.K..; Writing—Review and Editing, A.T.; Visualization, A.K. and A.S.

**Funding:** This research received no external funding.

**Conflicts of Interest:** The authors declare no conflict of interest.

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
