# Peer review of "Was Drought Really the Trigger Behind the Syrian Civil War in 2011?"

_water, doi:10.3390/w11081564_

Round 1

Reviewer 1 Report

The manuscript relies on an "eyeball analysis" of temporal data that is sometimes convincing, sometimes unconvincing, and which is certainly suitable for generating some interesting hypotheses but not sufficient for the level of confidence the authors express in their conclusions about what happened in Syria. Perhaps they are correct, the data are intriguing, but they are not sufficient to come to the new understanding they propose. In particular their interpretation of Figures 3 and 5 somewhat unconvincing, and the causal implications they draw from Figure 6 seem very weak considering how complicated a venture agriculture is. The authors have some very interesting ideas, but need to find stronger arguments for their conclusions than they present. 

Author Response

The reviewer is right in his statement that the manuscript “relies on an ‘eyeball analysis’”.  As presented in the manuscript, there are a considerable amount of articles, in scientific journals, as well as in popular publications, that explain the triggering forces before the Syrian civilian uprising in 2011.  Most, if not all, of these publications rely on climatic analyses and conclude that the pre-war droughts caused a massive population migration and thus, to the conflict onset.  In our mind, these studies are also relying on an ‘eyeball analysis’ while their cause and effect analysis had never been scientifically proven.  Our manuscript, however, rejects their prevailing hypothesis on the climatic-driven effect and presents an alternative one that is built upon much more solid evidences.  The strength of the current research is the strong link between several independent datasets, mostly based on earth observation data.  These are time series of (1) discharge in the Euphrates River; (2) vegetation spectral signals; (3) water level of major reservoirs; (4) yields of the representative crops for the rainfed agriculture in the winter (wheat) and that of the irrigated agriculture in the summer (cotton); and (5) transborder groundwater flow.  All these datasets led to the same conclusions.  Therefore, our synthesized research proves that the prevailing narrative, which blames new levels of drought for the country’s political instability, is inconsistent with actual environmental conditions.  On the contrary, complementary human-induced independent factors firmly point to reduced Euphrates flow as the most immediate cause of the 2011 Syrian agricultural collapse. 

Reviewer 2 Report

The paper claims that water shortages in Syria during the past decade were caused mainly by increased water abstraction in Turkey. The claim is based on remotely sensed data.

There is a number of questions that need to be asked concerning the methodology used:

The time series present seasonal mean values averaged over vast territories for a limited number of years.  Obviously, there are many missing values in the respective data sets. How does this impact the validity of the results? There is also a change of technology in the data collection during the period studied. How might this impact the reliability? It is also evident that the data-set is quite inhomogeneous as there has been a series of developments of large irrigation projects. How has this been taken into account?

In remote sensing applications, it is generally considered necessary to have some kind of “ground truth” to validate the relevance of the remotely sensed data. The paper does not give much information about conditions observed on the ground. There is no information about rainfall or land use.

We learn that the discharge measurements became inaccessible due to occupation of the adjacent land by the Islamic State. However, there is no information about how much land was occupied or what the occupation meant for the land use.

It would have been helpful to have some information about the yields/ha for typical crops during the time period considered in the analysis.

It seems that the trends of the time series have been estimated by means of visual inspection, rather than by means of some calculations. Such an approach is generally deemed to be rather fallible.

There is a mention of a water agreement between Syria and Turkey. However, there is no mention of the content, or whether it was breached by Turkey during the period studied.

Author Response

Comment: The time series present seasonal mean values averaged over vast territories for a limited number of years.  Obviously, there are many missing values in the respective data sets. How does this impact the validity of the results?

Reply: Several datasets had been analyzed and presented.  The monthly mean NDVI is based on long-term satellite data with no missing values; The water level in the reservoirs loses data from Feb. 2012 to Apr. 2013 for the Lake Ataturk and from Oct. 2010 to Apr. 2013.  However, despite the missing data, our chart shows that both reservoirs have a stable level until 2011 while the water level in the ‎Syrian lake gradually decreased after 2013.‎

Comment: There is also a change of technology in the data collection during the period studied. How might this impact the reliability?

Reply: Combining data from different space systems are common in remote sensing data analysis and should not affect reliability.  For example – for combining NOAA-AVHRR and MODIS datasets see Tucker et al. (2005), and for the multi-mission altimetry satellites, see Schwatke et al. (2015).

Tucker, C.J. Pinzon, J.E. Brown, M.E. et al. 2005. An extended AVHRR 8-km NDVI dataset compatible with MODIS and SPOT vegetation NDVI data.   International Journal of Remote Sensing26, 4485-4498.

Schwatke, C. Dettmering, D. Bosch, W. and Seitz, F. 2015. DAHITI – an innovative approach for estimating water level time series over inland waters using multi-mission satellite altimetry.  

Comment: It is also evident that the data-set is quite inhomogeneous as there has been a series of developments of large irrigation projects. How has this been taken into account?

Reply: The reviewer is right.  For taking into account the inhomogeneity of the areas due to agriculture expansion or change, the same constant areas, in Turkey and Syria, were analyzed throughout the years.

Comment: In remote sensing applications, it is generally considered necessary to have some kind of “ground truth” to validate the relevance of the remotely sensed data. The paper does not give much information about conditions observed on the ground.

Reply: The reviewer is right, but no ground truth activities were conducted on the ground since the area was not accessible during the war.  The project relies on the reliability of the space systems and the preprocessing procedures of the major space agencies (e.g., NASA, ESA).

Comment: There is no information about rainfall or land use.

Reply: We did not succeed to find reliable rainfall data for the years 2011-2015.  Land-use maps are usually produced from satellite images and since in our region there is only one land-use – agriculture, which is easily recognized from earth observation data, there is no need for looking for other sources. 

Comment: We learn that the discharge measurements became inaccessible due to the occupation of the adjacent land by the Islamic State. However, there is no information about how much land was occupied or what the occupation meant for land-use.

Reply: Between 2011 and 2015, the study area was controlled by the Islamic State.  We assume that the local population practiced agriculture as much as they could.

Comment: It would have been helpful to have some information about the yields/ha for typical crops during the time period considered in the analysis.

Reply: Data about typical crop productions, i.e., wheat and cotton as an example for winter rainfed and irrigated summer crop, exists in Figure 6.

Comment: It seems that the trends of the time series have been estimated by means of visual inspection, rather than by means of some calculations. Such an approach is generally deemed to be rather fallible.

Reply: We agree with the reviewer.  The difference between NDVI over the Turkish and Syrian territories and its trend along the years were quantified with the Cohen’s D Effect Size Index (D).  A new section was added to the Methodology and the analysis was explained in the Results including a new graph.

Comment: There is a mention of a water agreement between Syria and Turkey. However, there is no mention of the content, or whether it was breached by Turkey during the period studied.

Reply: The reviewer is right.  A summary of the 1987 and 2009 agreements between Turkey and Syria were added to the Introduction.

Round 2

Reviewer 1 Report

The insistence that your analysis disproves another correlation-based hypothesis oversteps what you achieve. You say  “Our synthesized research proves that the prevailing narrative, which blames new levels of drought for the country’s political instability, is inconsistent with actual environmental conditions. A range of quantifying factors, based on several independent sources, firmly point to reduced Euphrates flow as the most immediate cause of the 2011 Syrian agricultural collapse.” Actually you pose a very interesting alternative hypothesis, but are far from giving an argument that definitively distinguishes between them. This would be worth publishing if either it refrained from overstepping the strength of the argument and was phrased as presenting an alternative explanation, or if it was extended somehow to really allow one of the hypotheses to be definitively excluded.

Author Response

We have accepted the recommendation of the reviewer and rewritten the sentence that he found to be “Overstepping”.  We have even tapped the very language suggested.  In the Conclusions, the sentence now reads:   

"Our synthesized research offers an alternative explanation to the prevailing narrative, which blames new levels of drought for the country’s political instability, is inconsistent with actual environmental conditions."  

While we believe that our alternative view is indeed compelling, we are happy to have readers ‎form their own opinions.

Reviewer 2 Report

The paper has improved and become an easier read. However, I still have some concerns:

The authors show that he agricultural output has decreased in the Syrian part of the study area from 2011, while the Turkish part of the area, in contrast, has experienced an increased production. The authors claim that the dominant cause of this difference is a reduction of the flow of the Euphrates across the border.

It seems to me that the study region has mainly got its irrigation water from the underlying aquifer, an aquifer that has been systematically over-pumped during the past decades. The paper indicates that there are some 1500 un-registered wells in the area. Thus, a dwindling availability of groundwater is a contributing factor to lower agricultural output.

It is also stated that a part of the area was invaded by the Islamic State in 2011. There is however, no information about the size or location of the area concerned. Were any reservoir(s) affected?

The maps show a presence of some hydrometric stations within the area. However, it seems that no information from these stations has been used to support the thesis presented.

Finally, the Turkish plans for the development of their water resources have been well known for quite some time, and the paper even states that there an agreement concerning joint water planning for the region. The obvious question is then whether the Turks broke any agreement or if everything progressed according to the plans.  

It could have been nice to know a little about the evidence for their position that the other side of the debate have presented. Were they referring to Syria in general, or just to the region covered in this paper.

Author Response

Q1.  It seems to me that the study region has mainly got its irrigation water from the underlying aquifer, an aquifer that has been systematically over-pumped during the past decades. The paper indicates that there are some 1500 un-registered wells in the area. Thus, a dwindling availability of groundwater is a contributing factor to lower agricultural output.

A1.  The reviewer is basically correct as the diminishing amounts of ground water is one of the factors to the agriculture crisis in Syria.  This situation is explained in Subsection 3.3.  However, groundwater depletion was an additional factor to the rivers’ discharge.  In 1960, the mean annual discharge of the Euphrates River was about 1000 cub. m per sec while ground water discharge only 47 cub. m per sec.  When possible, farmers prefer to transfer water gravitationally in open channels rather than spend money and efforts on pumping water from below ground.

Q2.  It is also stated that a part of the area was invaded by the Islamic State in 2011. There is however, no information about the size or location of the area concerned. Were any reservoir(s) affected?

A2.  We added a territorial control map of the Syrian Civil War as of November 24, 2014.  The map shows the vast area controlled by the ISIS forces, specifically along the Euphrates River and Assad Lake.

Q3.  The maps show a presence of some hydrometric stations within the area. However, it seems that no information from these stations has been used to support the thesis presented.

A3..Indeed, there are other hydrometric stations along the three main tributaries of the Euphrates, Sajur, Balikh, ‎and Khabour.  However, their overall discharge was only 8% ‎of the main river.  We added to the text that their flows were also dramatically decreased through the years ‎

Q4. Finally, the Turkish plans for the development of their water resources have been well known for quite some time, and the paper even states that there an agreement concerning joint water planning for the region. The obvious question is then whether the Turks broke any agreement or if everything progressed according to the plans.  

A4. ‎Reviewer requests that we weigh in as to whether Turkey violated its water agreement with Syria – and so we have added the following sentence in the Conclusions: “This situation most likely happened since the Turkish government’ commitment to allowing a ‎minimum flow in the Euphrates at its Syrian border was violated when Turkey pursued its ‎ambitious ‎reservoir strategy.” 

Q5. It could have been nice to know a little about the evidence for their position that the other side of the debate have presented. Were they referring to Syria in general, or just to the region covered in this paper.

A5. We cannot say that the other papers are related exactly to the same polygons of the current research since each paper has its perspective, specific objectives, and research areas.  Chatel (reference no. 1) and Femia and Werrell (reference nos. 2 and 3) are referring to climate change in the entire Syrian territory.  Fetzek and Mazo (reference no. 4) analyzed the Syrian conflict in general, but states the “More than 70% of the country’s freshwater resources come from transborder flows, the bulk from Turkey via the Euphrates River”.  Gleick (reference no. 5) generally refer to the same area as in the current paper and mentioned the tension between Turkey and Syria due to the substantial